# Electromyographic analysis of the stomatognathic system of children with Molar-incisor hypomineralization

**Milena Rodrigues Carvalho**[1], **Simone Cecilio Hallak Regalo**[1,2], **Selma Siéssere**[1,2], **Lígia Maria Napolitano Gonçalves**[1], **Francisco Wanderley Garcia de Paula-Silva**[1], **Fernanda Vicioni-Marques**[1], **Paulo Nelson-Filho**[1], **Paulo Batista de Vasconcelos**[2], **Fabrício Kitazono de Carvalho**[1]*, **Alexandra Mussolino de Queiroz**[1]

1 Ribeirão Preto School of Dentistry, University of São Paulo, São Paulo, Brazil, 2 National Institute for Translational Medicine (INCT-TM), Ribeirão Preto, Brazil

* fabricio_kc@forp.usp.br

**Data Availability Statement:** All relevant data are within the paper and its Supporting Information files.

## Abstract

### Introduction

Molar-incisor hypomineralization (MIH) is a qualitative defect of dental enamel that affects one or more permanent first molars, with or without involvement of the incisor teeth. This condition leads to challenges to dental care and treatment planning.

### Aim

Based on the hypothesis that children who have MIH possibly present alterations in postural and masticatory activities and considering the absence of studies investigating these parameters, the present study evaluated the functionality of the stomatognathic system considering the mentioned aspects.

### Materials

The comparison of individuals with (MIHG; n = 32) and without MIH (CG; n = 32) was evaluated by electromyographic activity of the masseter and temporal muscles (right and left), as well as evaluation of the masticatory cycles during habitual mastication.

### Results

MIHG showed muscle hyperactivity in postural and dynamic conditions compared to the CG; higher electromyographic values for MIHG when compared to CG in the following postural conditions: at rest for the right temporal (p = 0.00) and left temporal muscles (p = 0.03); in the protrusion to the right temporal muscle (p = 0.02); in the right laterality for the right masseter (p = 0.00) and left temporal muscles (p = 0.01); in the left laterality for the right masseter (p = 0.03) and left temporal (p = 0.04) muscles. In dynamic conditions with consistent food, significance was observed for the left temporal (p = 0.01); and with soft food for the right (p = 0.01) and left temporal muscles (p = 0.04).

**Funding:** This research was funded by the National Council for Scientific and Technological Development – CNPq (Process: 405914/2021-0). The funders had no role in study design, data collection and analysis, decision to publish, or preparation of the manuscript.

**Competing interests:** The authors have declared that no competing interests exist.

## Conclusions

Children with MIH seem to have impaired functionality of the stomatognathic system. Children with MIH have alterations in the stomatognathic system.

## Introduction

Molar-incisor hypomineralization (MIH) was first described in Sweden [1] in the late 1970s. Nonetheless, the term MIH was introduced only in 2001 to report qualitative developmental dental enamel defects affecting one or more permanent first molars, with or without involvement of permanent incisors. Clinically, MIH is characterized by the presence of demarcated white or yellowish-brown opacity in at least one permanent first molar, which may asymmetrically and heterogeneously affect other teeth [2,3]. Currently, MIH etiology is considered multifactorial, with systemic factors affecting genetically predisposed individuals [4–6].

MIH has a global mean prevalence of 14.2% [7] varying greatly between continents, with 18.0% in America, 16.3% in Australia, 14.3% in Europe, 13.0% in Asia, and 10.9% in Africa. When analyzed by country, MIH has a prevalence of 8.1% in India, 10.5% in Germany, 16.0% in Finland, 21.1% in Spain, and 17.1% in Iran. In Brazil, MIH has an estimated mean prevalence of 20% [7–11].

Hypomineralized enamel has lower hardness and elastic modulus than normal enamel [12–17]. Due to limited mechanical properties, hypomineralized enamel is more vulnerable to posteruptive breakdown (PEB), which can occur suddenly during severe MIH making the structure seem clinically undeveloped. PEB has a higher incidence in permanent first molars than in permanent incisors due to greater masticatory force in the molar region [3,17,18].

A dentition within the normal range has a tendency toward equal bilateral dental contacts around the sagittal axis, and the stress center for anteroposterior dental contacts is symmetrical and bilateral in the region of the first molars [19,20].

Surface electromyography allows the exact study of the interpretation of normal and pathological conditions associated with muscle functions [21–24]. Thus, this gold standard methodology has been used in the study of myalgias, occlusal changes and also during the diagnosis and treatment of neuromuscular diseases, allowing to explain the biomechanical and functional responses of muscle activities at rest and in other dynamic conditions such as chewing [25–30].

Numerous studies have been carried out to demonstrate how muscle activities can interfere with the performance of the entire stomatognathic system, such as the severity of signs and symptoms of temporomandibular disorders (TMD) in children is related to both morphological and functional changes in the stomatognathic system, demonstrating the relevance of analyzing electromyographic activity [31]; age as a predisposing factor in changing the electromyographic fatigue threshold in the activities of the masseter and temporal muscles as healthy individuals age [32]; children with borderline orthodontic treatment needs had functional disorders of the stomatognathic system [33]; the influence of sleep bruxism severity on masticatory efficiency [34]; in addition to other analyzes such as breastfeeding, the introduction of food and the use of pacifiers in the analysis of the stomatognathic system of children [35]; individuals with TMD used an occlusal splint, superior or inferior, in order to verify a reduction in muscle pain, but after comparing the electromyography exams performed before and after 6 months of splint use, it was noted that there was no significant change in pain. myofascial, suggesting the addition of other treatments concomitantly with the use of the

splint, such as psychological counseling, analgesics and oral rehabilitation [36]; faced with a COVID-19 pandemic scenario, the telerehabilitation of a woman of approximately 43 years who had chronic peripheral facial paralysis of the frontal and orbicularis oris muscles was carried out, in the electromyographic analysis, an improvement in the symmetry of the muscles in question was observed [37]; individuals with obstructive sleep apnea were evaluated in order to verify the number and type of mandibular muscles that would be involved to validate sleep bruxism, thus it was found that there was no necessary score for rhythmic masticatory muscle activity against sleep apnea[38].

Thus, although there are numerous studies on muscle assessment through surface electromyography, studying the possible interference of the HMI in the stomatognathic system made this study original and extremely relevant since this topic is not addressed in the HMI and evaluation relationship. muscle.

This study evaluated the functionality of the stomatognathic system based on the hypothesis that children with MIH present postural and masticatory activity changes and considering the lack of literature on these parameters. Thus, the objective of this study was to conduct a comparative analysis of the electromyographic (EMG) activity of the masseter and temporal muscles (right and left), as well as to evaluate habitual masticatory cycles in individuals with and without MIH.

## Material and methods

### Study design and sample selection

A comparative cross-sectional observational study was conducted to analyze EMG activity patterns in two groups, with and without MIH. The research hypothesis was that participants with MIH presented changes in all analyzed variables. The null hypothesis would show no difference between the two analyzed groups. This project was approved by the Research Ethics Committee of the Ribeirão Preto School of Dentistry, University of São Paulo (FORP/USP). All participants signed the Free and Informed Consent Form to participate in this study. Before the beginning of the study, the research project was approved by the Research Ethics Committee in accordance with Resolutions 466/12 and Complementary, complying with the Helsinke Resolution. Consent is in writing. All Participants provided written informed consent to participate in this study. When the participants were minors (18 years old), their parents signed the term, as long as the children agreed to collaborate and participate in the study. The research project was approved by the Research Ethics Committee in accordance with the resolutions of the CEP/CONEP System that govern ethical research regulations in Brazil.

For calibration, the examiner (MRC) underwent extensive training, which included theoretical activities, discussion with other authors regarding the criteria described by Ghanim et al. (2015) [39], and a calibration exercise with the standard examiner (FKC) using 30 clinical photos, with the examiner reaching a Kappa index > 0.8.

We recruited children aged between six and 12 years of both sexes, referred to the FORP/USP Dental Enamel Clinic by different municipal schools in Ribeirão Preto and adjacent region.

The inclusion criteria for the MIH group were at least one first permanent molar presenting indices compatible with MIH (codes 2, 21, 22, and 3 of the criteria by Ghanim et al., 2015) [39]. The exclusion criteria were treatment with muscle relaxants, corticosteroids, and immunosuppressants; a medical history of surgical intervention less than 12 months before the beginning of the study; use of anti-inflammatory drugs and/or analgesics that could interfere with neuromuscular physiology; oral and/or facial surgical interventions with facial and/or

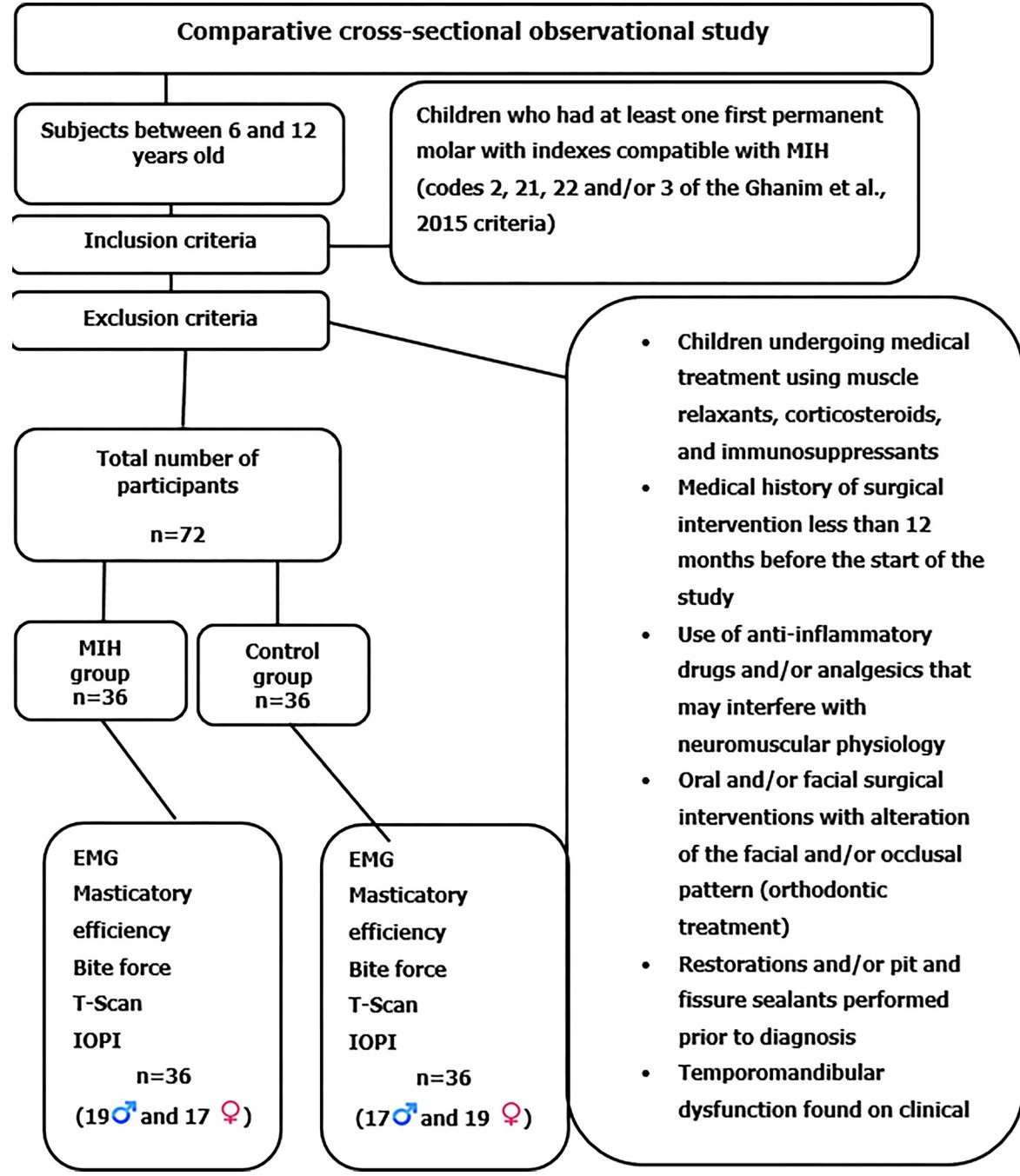

**Fig 1. Diagram of the study population sampling.**

occlusal pattern changes, including orthodontic treatments; pit and fissure sealants and/or restorations prior to diagnosis and/or temporomandibular disorders (TMD) upon clinical examination (Fig 1).

Subjects in the groups were age-sex-matched. Sample calculation determined the required number of subjects for each group with a significance level of 5% and a test power of 95% based on the results published by Regalo et al., 2008 [40], resulting in 36 subjects per group (Fig 1).

The MIHG comprised children with at least one permanent first molar with demarcated opacities (code 2, 21, and 22 of the criteria by Ghanim et al., 2015) [39], PEB (code 3 of the criteria by Ghanim et al., 2015) [39], with no restoration or pit and fissure sealant in this element and no other permanent first molar with caries, restoration, or pit and fissure sealant.

The CG comprised children presenting no teeth with MIH and no restoration, pit, and fissure sealant, or caries in any permanent first molar.

## Electromyographic analysis

Surface electromyography captured EMG signals using a 16-channel Trigno TM Wireless System electromyograph (20–450 Hz; common-mode rejection ratio CMRR of 80 dB, input impedance higher than 1000 X, Delsys Inc., Boston, MA, USA), Trigno sensors (50 mm$^2$ area; USA patent 6480731, 6238338; European Patent EP 1070479), formed by four parallel bars (Ag-AgCl), two references, and two active surface electrodes with a 10 mm distance between electrodes. The obtained EMG signals were digitized, amplified (gain = 300), and sampled at 4 kHz with a 16-bit resolution A/D system (EMG Works acquisition software, Delsys Inc., Boston, MA, USA). Sensor centers were distanced at least 20 mm from one another, and total channel noise was < 0.45 1V pk-pk.

The environment had to be calm and silent to correctly record EMG activities in postural and dynamic conditions. The children were placed in a comfortable chair, in an upright posture, and an accessory wooden bench/ladder was provided to support their feet so that the soles were parallel to the ground and their hands were resting on their thighs. The patient's head remained positioned in the Frankfort horizontal plane, parallel to the ground. This postural condition was standardized for all tests in the study.

All sensors were placed on the participants by a single specialized and trained technician throughout the study to ensure they were systematically positioned according to the Surface ElectroMyoGraphy for the Non-Invasive Assessment of Muscles (SENIAM) norms. Prior to placing the sensors, the patient's skin was cleaned with alcohol to eliminate possible grease or pollution residues. Each muscle evaluated was digitally palpated to ensure its correct location, and the participant was asked to perform maximum voluntary contraction for proper electrode positioning [41,42].

The patient's postural EMG activity was assessed with the mandible in the following postural clinical conditions: rest (4s), right and left laterality with dental contact (5s), protrusion with dental contact (5s), maximum dental clenching with and without Parafilm M® (4s) (inert material), and measurements recorded in Root Mean Square (RMS). Maximum dental clenching with Parafilm M® was used to normalize all data collected from surface EMG tests, according to SENIAM norms. Parafilm M® is an inert material consisting of a folded paraffin sheet (Parafilm M®, Pechinery Plastic Packaging, Batavia, IL, USA) (18 x 17 x 4 mm, 245 mg), which was placed between the occlusal surfaces of the superior and inferior first molars in the right and left sides of the dental arch.

EMG activity was analyzed using masticatory muscle records according to the protocol shown in Table 1. The patient's habitual and non-habitual mastication was dynamically

**Table 1. Electromyographic protocol of masticatory activity.**

| Muscles | Clinical conditions | Normalization |
|---|---|---|
| Masseters and temporal | Soft food mastication (10s) (chocolate covered peanuts) | CVM (normalization factor) (4s) |
| Masseters and temporal | Consistent food mastication (10s) (chocolate covered wafer) | CVM (normalization factor) (4s) |

assessed by the efficiency of the masticatory cycles and analyzed through the linear EMG signal envelope of the mastication cycles using the masseter and temporal muscles, bilaterally, with the values recorded in microvolts/s ($\mu$V) [43].

During habitual chewing, EMG signals were collected from the mastication cycles of the masseter and temporal muscles, bilaterally, with soft (chocolate covered wafer, Bis-Lacta®) and consistent food (chocolate covered peanuts, M&M's) for a period of 10s each. Before this test, the patients were informed about the type of food to be chewed, noting that they did not need to swallow it as they would have the option of discarding the food in an appropriate container if they did not want to consume it (Table 2).

The data were acquired 5–10s after the beginning of the mastication cycle, as the initial five seconds were eliminated because the first cycles of the masticatory process present significant mandibular movement pattern variation [44].

The raw EMG signal of the analyzed muscles (masseter and temporalis, right and left sides) was used to derive EMG amplitude values obtained by calculating the RMS. The masticatory cycle function was analyzed by the linear envelope integral of the EMG signal in microvolts/s.

The data obtained from EMG activities were normalized by maximal voluntary contraction with Parafilm M® in static (rest, protrusion, right laterality, left laterality) and dynamic conditions (chewing with consistent and soft food) and underwent the Kolmogorov-Smirnov normality test. Normal data distribution was observed for the MIHG and CG (without MIH). Significance level was set at 5%.

## Results

The study included 72 children aged between six and 12 years, divided into the MIH group (MIHG, n = 36) and a control group without MIH (CG, n = 36). Table 3 shows the subject-to-subject matching of the MIHG and CG. In the MIHG, participants had a mean age of 8.3056 (standard deviation (SD) 1.6530) years, and in the CG the mean age was 8.5556 (SD 0.84327). The MIHG had a mean body mass index (BMI) of 19.9638 kg/m$^2$ (SD 23.901) and the CG of 18.8119 kg/m$^2$ (SD 28.1760).

The analysis of the mandibular rest condition showed a statistically significant difference between the two groups for the right and left temporal muscles ($p \leq 0.05$). According to the analysis of normalized electromyographic means, participants belonging to the MIHG presented higher values for the left masseter and right and left temporal muscles (Table 3 and Fig 2).

In the maximum protrusion condition, there was a statistically significant difference ($p \leq 0.05$) between the two groups for the right temporal muscle. The analysis of normalized electromyographic means of participants belonging to the MIHG presented higher electromyographic values for all muscles evaluated (Table 4 and Fig 3).

The analysis of the right laterality condition showed a statistically significant difference between the two groups for the right masseter and left temporal muscles ($p \leq 0.05$). According to the analysis of normalized electromyographic means, participants belonging to the MIHG presented higher values for the right masseter and right and left temporal muscles (Table 5 and Fig 4).

In the left laterality condition, there was a statistically significant difference ($p \leq 0.05$) between the two groups for the right masseter and left temporal muscles. The analysis of normalized electromyographic means of participants belonging to the MIHG presented higher values for all muscles evaluated (Table 6 and Fig 5).

The analysis of habitual mastication of consistent food by the linear EMG signal envelope of the mastication cycles showed a statistically significant difference ($p \leq 0.05$) between the

**Table 2. Subject-to-subject matching of the MIH group (MIHG) and the control group (CG).**

| Participants | Sex | | Age | | BMI | |
|:---:|:---:|:---:|:---:|:---:|:---:|:---:|
| (n = 72) | MIHG | CG | MIHG | CG | MIHG | CG |
| 1 | F | F | 11 | 10 | 27.83 | 23.37 |
| 2 | M | M | 7 | 9 | 22.91 | 18.59 |
| 3 | F | F | 9 | 9 | 31.20 | 30.80 |
| 4 | F | F | 8 | 8 | 14.08 | 13.25 |
| 5 | M | M | 10 | 10 | 15.87 | 11.25 |
| 6 | F | F | 9 | 9 | 29.54 | 25.45 |
| 7 | M | M | 8 | 8 | 17.82 | 17.58 |
| 8 | M | M | 10 | 9 | 16.10 | 18.59 |
| 9 | F | F | 9 | 10 | 13.77 | 13.79 |
| 10 | F | F | 10 | 9 | 26.76 | 23.78 |
| 11 | F | F | 7 | 8 | 22.57 | 21.97 |
| 12 | F | F | 9 | 8 | 21.20 | 21.97 |
| 13 | F | F | 7 | 7 | 15.30 | 15.73 |
| 14 | M | M | 7 | 8 | 16.43 | 15.96 |
| 15 | M | M | 11 | 9 | 27.28 | 18.59 |
| 16 | F | F | 9 | 9 | 26.27 | 25.45 |
| 17 | M | M | 12 | 9 | 18.03 | 18.59 |
| 18 | F | F | 6 | 7 | 17.33 | 15.73 |
| 19 | M | M | 6 | 8 | 16.18 | 15.62 |
| 20 | M | M | 7 | 8 | 16.69 | 17.33 |
| 21 | F | F | 8 | 9 | 18.07 | 18.76 |
| 22 | M | M | 7 | 8 | 16.12 | 17.33 |
| 23 | M | M | 7 | 10 | 12.21 | 11.25 |
| 24 | M | F | 10 | 9 | 21.74 | 18.64 |
| 25 | F | F | 8 | 8 | 17.50 | 17.28 |
| 26 | F | F | 10 | 10 | 18.00 | 18.06 |
| 27 | M | M | 6 | 8 | 20.80 | 15.96 |
| 28 | F | M | 6 | 9 | 20.20 | 18.59 |
| 29 | M | F | 7 | 8 | 16.00 | 21.97 |
| 30 | M | M | 7 | 8 | 21.70 | 15.96 |
| 31 | F | F | 7 | 8 | 18.20 | 21.97 |
| 32 | F | F | 11 | 7 | 37.10 | 15.73 |
| 33 | M | F | 8 | 9 | 12.80 | 30.80 |
| 34 | M | M | 10 | 8 | 16.10 | 15.62 |
| 35 | M | M | 7 | 8 | 23.30 | 17.33 |
| 36 | M | M | 8 | 9 | 15.70 | 18.59 |

EMG, electromyographic; MIH, Molar-incisor hypomineralization; CG, control group; MIHG, MIH group; F, female; M, male; BMI, body mass index.

two groups for the left temporal muscle. The MIHG presented higher electromyographic means for all muscles studied (Table 7 and Fig 6).

The analysis of habitual mastication of soft food by the linear EMG signal envelope of the mastication cycles showed a statistically significant difference (p ≤ 0.05) between the two groups for the left temporal muscle. The MIHG presented higher electromyographic means for all muscles studied (Table 8 and Fig 7).

**Table 3. Mean values, standard error, and statistical significance (p ≤ 0.05) of the normalized electromyographic data in the mandibular rest condition, for each muscle evaluated, in the MIH group (MIHG) and control group (CG).**

| Muscle | Group | EMG Means | Standard Error (±) | Significance (*) |
|---|---|---|---|---|
| Right masseter | MIHG | 0.11 | 0.02 | 0.97 |
| | CG | 0.11 | 0.02 | |
| Left masseter | MIHG | 0.13 | 0.02 | 0.23 |
| | CG | 0.09 | 0.01 | |
| Right temporal | MIHG | 0.59 | 0.06 | **0.00**** |
| | CG | 0.13 | 0.01 | |
| Left temporal | MIHG | 0.24 | 0.05 | **0.03**** |
| | CG | 0.11 | 0.01 | |

EMG, electromyographic; MIH, Molar-incisor hypomineralization; CG, control group; MIHG, MIH group. Significance < 0.05 = *; Significance < 0.01 = **; Significance < 0.1 = ***.

## Discussion

People with MIH present clinical conditions that directly affect their health and negatively impact their quality of life [27]. These conditions include dental hypersensitivity, tooth enamelpost eruptive breakdown (PEB), rapid progression of carious lesions in newly erupted first permanent molars, anesthetic difficulties, impaired aesthetics, in addition to fear and anxiety

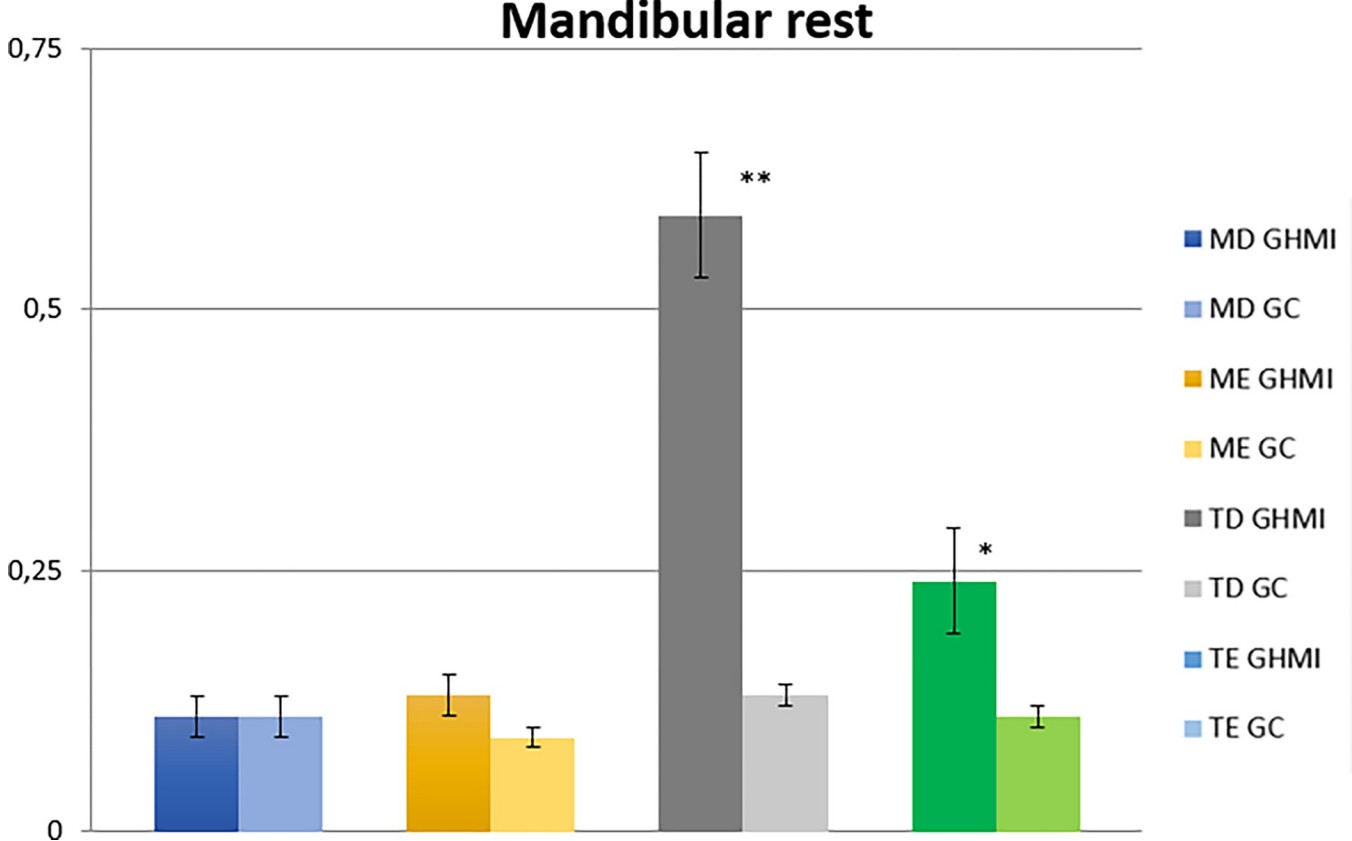

**Fig 2. Normalized electromyographic means (RMS) in the mandibular rest condition for the right masseter (RM), left masseter (LM), right temporal (RT) and left temporal (LT) muscles, in the MIH (MIHG) and control (CG) groups.** MIH, Molar-incisor hypomineralization.

**Table 4. Mean values, standard error, and statistical significance (p ≤ 0.05) of the normalized electromyographic data in the maximum protrusion condition, for each muscle evaluated, in the MIH group (MIHG) and control group (CG).**

| Muscle | Group | EMG Means | Standard Error (±) | Significance (*) |
|--------|-------|-----------|--------------------|------------------|
| Right masseter | MIHG | 0.78 | 0.24 | 0.62 |
| | CG | 0.63 | 0.16 | |
| Left masseter | MIHG | 0.76 | 0.22 | 0.45 |
| | CG | 0.56 | 0.13 | |
| Right temporal | MIHG | 0.36 | 0.07 | **0.02**** |
| | CG | 0.18 | 0.03 | |
| Left temporal | MIHG | 0.27 | 0.06 | 0.11 |
| | CG | 0.16 | 0.03 | |

EMG, electromyographic; MIH, Molar-incisor hypomineralization; CG, control group; MIHG, MIH group. Significance < 0.05 = *; Significance < 0.01 = **; Significance < 0.1 = ***.

during dental care [3,16,17,28,29]. Certain clinical conditions, such as increased sensitivity, PEB and predisposition to the development of carious lesions in the affected teeth [6,45] could cause masticatory difficulties and, be associated with changes in the functions of the muscles involved in this process.

PEB in teeth with MIH may be related to lower hardness and modulus of elasticity of hypomineralized enamel [11–16]. Thus, the mechanical properties are limited for individuals with this condition. The incidence of PEB is higher in permanent first molars than in permanent incisors, since masticatory forces are greater in the molar region [3,16,17]. Therefore, this is a critical point between the functional relationships of the stomatognathic system and MIH, i.e.,

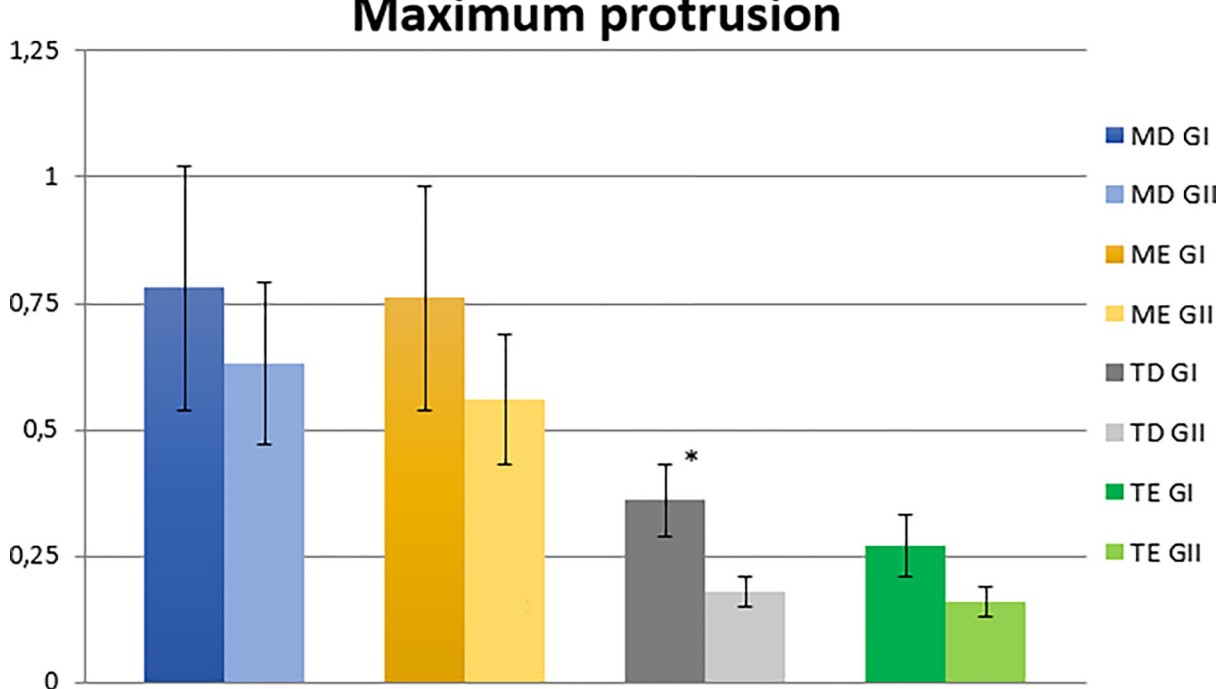

**Fig 3. Normalized electromyographic means (RMS) in the maximum protrusion condition for the right masseter (RM), left masseter (LM), right temporal (RT) and left temporal (LT) muscles, in the MIH (MIHG) and control (CG) groups.** MIH, Molar-incisor hypomineralization.

**Table 5. Mean values, standard error, and statistical significance (p ≤ 0.05) of the normalized electromyographic data in the right laterality condition, for each muscle evaluated, in the MIH group (MIHG) and control group (CG).**

| Muscle | Group | EMG Means | Standard Error (±) | Significance (*) |
|--------|-------|-----------|---------------------|------------------|
| Right masseter | MIHG | 1.65 | 0.45 | **0.00**** |
| | CG | 0.26 | 0.08 | |
| Left masseter | MIHG | 0.30 | 0.06 | 0.20 |
| | CG | 0.52 | 0.15 | |
| Right temporal | MIHG | 0.30 | 0.08 | 0.80 |
| | CG | 0.28 | 0.07 | |
| Left temporal | MIHG | 0.21 | 0.03 | **0.01**** |
| | CG | 0.10 | 0.01 | |

EMG, electromyographic; MIH, Molar-incisor hypomineralization; CG, control group; MIHG, MIH group. Significance < 0.05 = *; Significance < 0.01 = **; Significance < 0.1 = ***.

the PEB of dental enamel could lead to masticatory dysfunctions that could affect the balance of the stomatognathic system.

The composition of the stomatognathic system is formed by bone structures, teeth, muscles, joints, glands, and vascular, lymphatic, and nervous systems, which work harmoniously during mandibular postural conditions and mastication [46]. However, in children with MIH, it was unknown whether enamel defects in permanent molars could affect this harmony. Thus, we conducted the present study, with different internationally recognized methodologies, to answer this question and accurately assess the masticatory muscles of individuals with MIH.

Surface electromyography is the gold standard to assess muscle activity of any muscle of interest. In the present study, the masseter and temporal muscles of the right and left sides

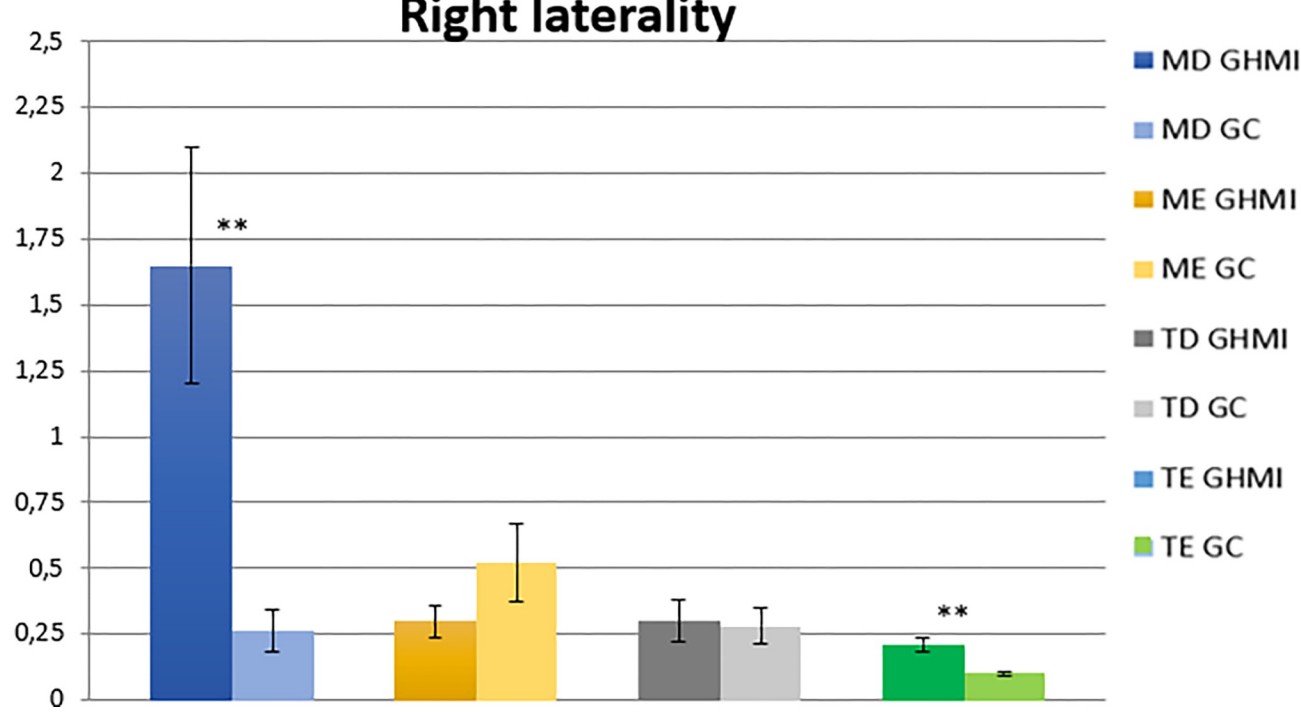

**Fig 4. Normalized electromyographic means (RMS) in the right laterality condition.**

**Table 6. Mean values, standard error, and statistical significance (p ≤ 0.05) of the normalized electromyographic data in the left laterality condition, for each muscle evaluated, in the MIH group (MIHG) and control group (CG).**

| Muscle | Group | EMG Means | Standard Error (±) | Significance (*) |
|---|---|---|---|---|
| Right masseter | MIHG | 1.62 | 0.59 | **0.03**** |
| | CG | 0.31 | 0.06 | |
| Left masseter | MIHG | 0.27 | 0.05 | 0.19 |
| | CG | 0.19 | 0.02 | |
| Right temporal | MIHG | 0.26 | 0.06 | 0.06 |
| | CG | 0.14 | 0.02 | |
| Left temporal | MIHG | 0.29 | 0.04 | **0.04**** |
| | CG | 0.18 | 0.01 | |

EMG, electromyographic; MIH, Molar-incisor hypomineralization; CG, control group; MIHG, MIH group. Significance < 0.05 = *; Significance < 0.01 = **; Significance < 0.1 = ***.

were evaluated, in the following postural conditions (Table 9): mandibular rest, maximum protrusion with dental contact, maximum right and left laterality with dental contact, and tooth clenching in maximum voluntary contraction with Parafilm M®. The latter condition was used for the normalization of all these data.

In the analysis of mandibular rest, the left masseter, right and left temporal muscles had higher values for electromyographic activities in the group with MIH (MIHG). This fact

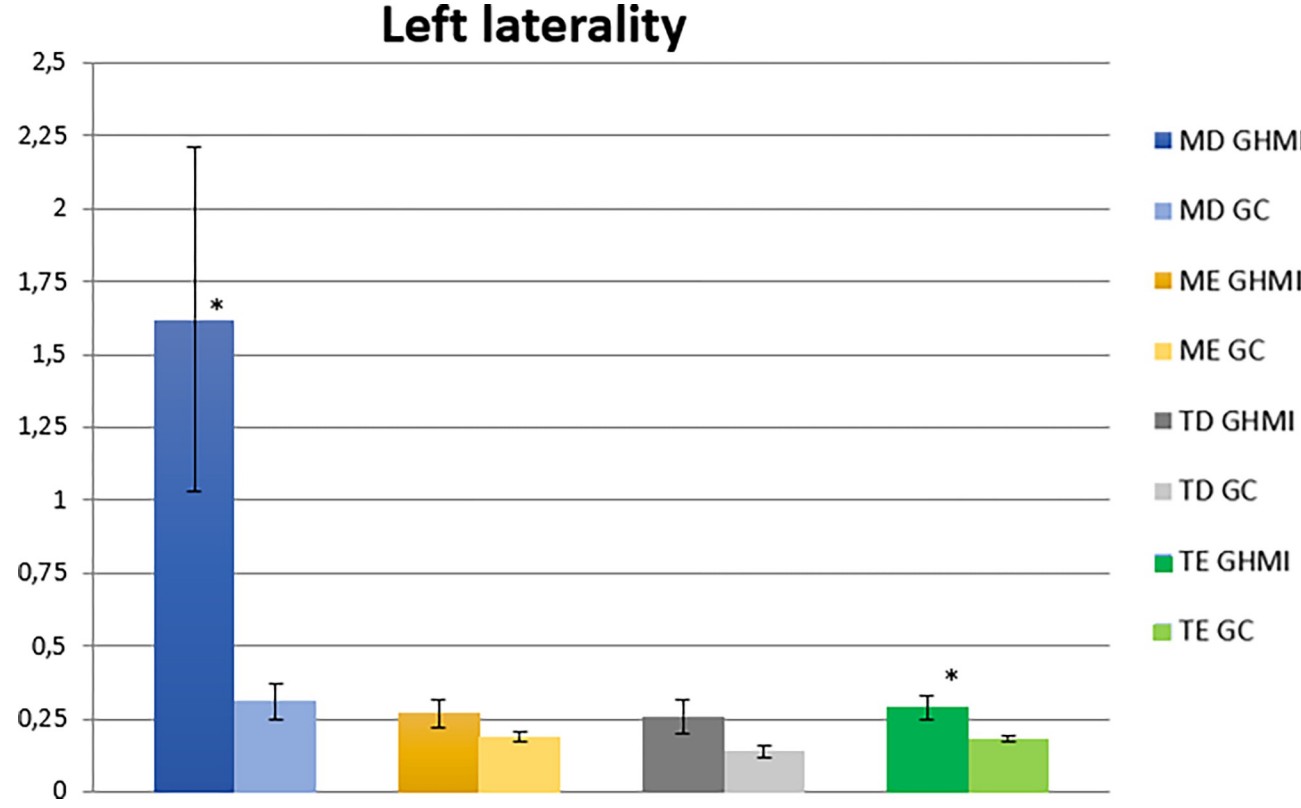

**Fig 5. Normalized electromyographic means (RMS) in the right laterality condition for the right masseter (RM), left masseter (LM), right temporal (RT) and left temporal (LT) muscles, in the MIH (MIHG) and control (CG) groups.** MIH, Molar-incisor hypomineralization.

**Table 7. Mean values, standard error, and statistical significance (p ≤ 0.05) of the normalized electromyographic data in the habitual mastication of consistent food condition, for each muscle evaluated, in the MIH group (MIHG) and control group (CG).**

| Muscle | Group | EMG Means | Standard Error (±) | Significance (*) |
|---|---|---|---|---|
| Right masseter | MIHG | 1.06 | 0.16 | 0.56 |
| | CG | 0.92 | 0.16 | |
| Left masseter | MIHG | 1.09 | 0.15 | 0.56 |
| | CG | 0.96 | 0.15 | |
| Right temporal | MIHG | 0.97 | 0.17 | 0.13 |
| | CG | 0.69 | 0.04 | |
| Left temporal | MIHG | 0.94 | 0.15 | **0.01**\*\* |
| | CG | 0.55 | 0.04 | |

EMG, electromyographic; MIH, Molar-incisor hypomineralization; CG, control group; MIHG, MIH group. Significance < 0.05 = *; Significance < 0.01 = **; Significance < 0.1 = ***.

demonstrates an imbalance in the resting condition in the MIHG, because in the normal pattern the mastication muscles should present minimal or nonexistent activities [47]. The mandibular balance is maintained by the action of the viscoelasticity of the masticatory muscles, the proprioceptive action of the ligaments, tendons, joint capsule, and atmospheric pressure [48–50]. Our results corroborate those of other studies that demonstrated the presence of electrical activity in skeletal striated muscles in situations of continuous stress [51–54]. Children who have MIH usually have increased sensitivity in the affected teeth [27], which are more sensitive to external stimuli such as cold, heat, toothbrush friction, and acidic foods. Furthermore, people who have teeth with MIH and enamel PEB have more pain sensitivity when compared to those who have only mild enamel defects [2,55]. Thus, faced with this scenario, there is a decrease in the patient's quality of life, causing increased stress, as reported in the literature.

In the protrusion condition, children with MIH showed higher electromyographic values compared to children without MIH for all muscles evaluated (masseter and temporal, right

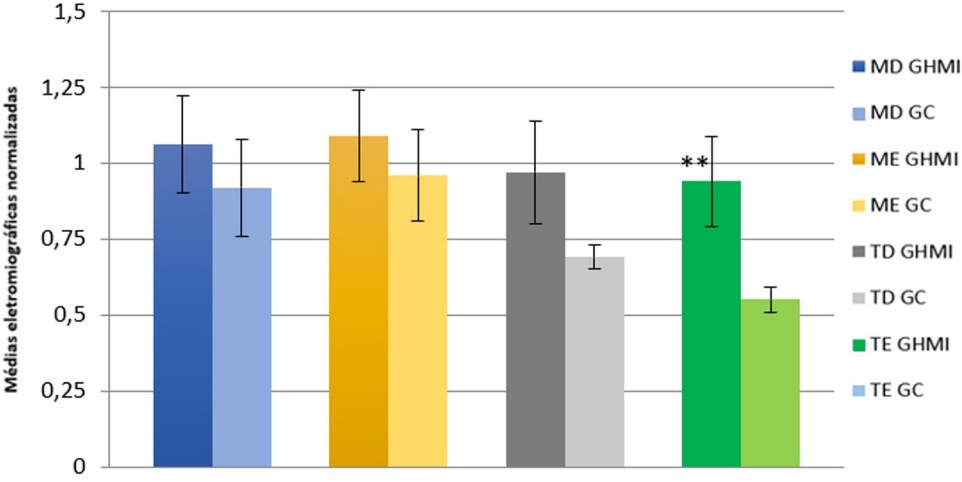

**Fig 6. Normalized electromyographic means (RMS) in the habitual mastication of consistent food condition for the right masseter (RM), left masseter (LM), right temporal (RT) and left temporal (LT) muscles, in the MIH (MIHG) and control (CG) groups.** MIH, Molar-incisor hypomineralization.

**Table 8. Mean values, standard error, and statistical significance (p ≤ 0.05) of the normalized electromyographic data in the habitual mastication of soft food condition, for each muscle evaluated, in the MIH Group (MIHG) and Control Group (CG).**

| Muscle | Group | EMG Means | Standard Error (±) | Significance (*) |
|---|---|---|---|---|
| Right masseter | MIHG | 0.89 | 0.13 | 0.89 |
|  | CG | 0.87 | 0.08 |  |
| Left masseter | MIHG | 1.14 | 0.21 | 0.29 |
|  | CG | 0.86 | 0.14 |  |
| Right temporal | MIHG | 1.19 | 0.22 | **0.01**** |
|  | CG | 0.60 | 0.05 |  |
| Left temporal | MIHG | 0.90 | 0.13 | **0.00**** |
|  | CG | 0.49 | 0.03 |  |

EMG, electromyographic; MIH, Molar-incisor hypomineralization; CG, control group; MIHG, MIH group. Significance < 0.05 = *; Significance < 0.01 = **; Significance < 0.1 = ***.

and left). That is, the higher electromyographic values in children with MIH mean they have muscle hyperactivity, causing imbalance in the stomatognathic system. Our results also showed that the masseter muscles were more active than the temporal muscles in this postural condition. These results agree with the movement normality patterns in this postural condition. In maximum protrusion with dental contact, the masseter muscles have greater activation, required by the muscular behavior itself, to remain in the desired position, when compared to the temporal muscles [56–59].

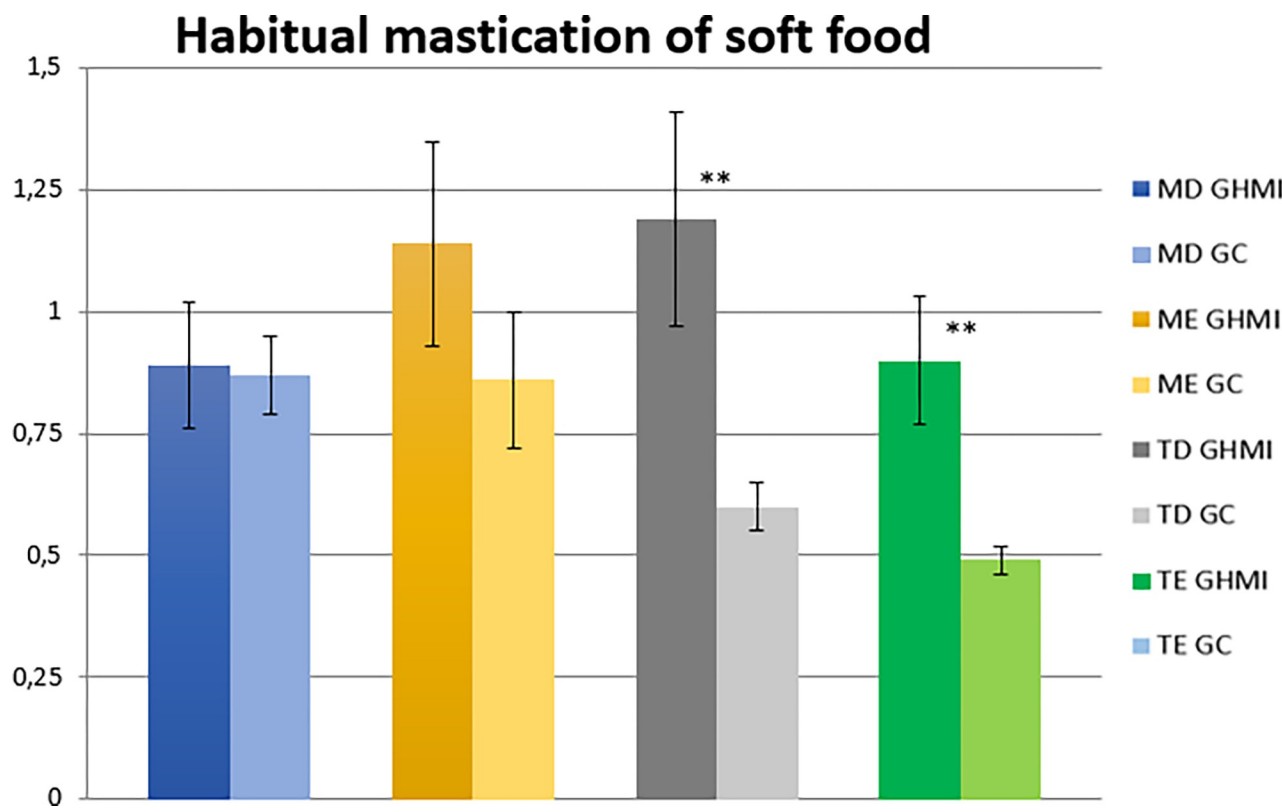

**Fig 7. Normalized electromyographic means (RMS) in the habitual mastication of soft food condition for the right masseter (RM), left masseter (LM), right temporal (RT) and left temporal (LT) muscles, in the MIH (MIHG) and control (CG) groups.** MIH, Molar-incisor hypomineralization.

**Table 9. Electromyographic protocol of postural conditions.**

| Muscles | Clinical conditions | Normalization |
|---|---|---|
| Masseters and temporal | Rest (4s) Protrusion (5s) Right laterality (5s) Left laterality (10s) Dental clenching on Parafilm M®(4s) | CVM (normalization factor) (4s) |

In the analysis of the right and left lateralities in children with MIH, we observed greater electromyographic activities for practically all the analyzed muscles (masseter and temporal, right and left), demonstrating again a muscular hyperactivity in this type of movement, except for the left masseter muscle, on the right laterality condition.

In the movement of maximal right and left laterality, with dental contact, the neuroanatomical muscle activation pattern demonstrates that greater electromyographic activity should be recorded in the temporal muscle of the ipsilateral side of the mandible (working side). Meanwhile, for the masseter muscle a greater contralateral activity is expected [56,60,61]. The present study observed this activation pattern, both in the right and left laterality in the CG and in the left laterality in the MIHG. Thus, we identified that the right side has hyperactivity in the right masseter and right and left temporal muscles, suggesting an imbalance on this side, which may be associated with the side with greater MIH involvement. However, this assumption requires further analyses. The hardness and elastic modulus of the enamel of hypomineralized teeth are lower than those found in normal enamel, establishing limited mechanical properties, making it more susceptible to PEB [11–16], and more sensitive. This causes discomfort in the movement, suggesting that the individual uses the contralateral side more often during mouth movement. This fact is demonstrated by the statistical significance observed in the right masseter and left temporal muscles in the right laterality condition, so that the exact opposite would happen in this movement, that is, greater activity for the left masseter and right temporal muscles during this movement.

In the electromyographic activity, the following dynamic conditions of mastication with consistent and soft food were observed. In habitual mastication, both with consistent (chocolate covered peanuts, M&M's) and soft food (chocolate covered wafer, Bis-Lacta®), the MIHG displayed higher electromyographic means for all muscles studied. Statistical significance was observed for the left temporal in mastication with consistent and with soft food, and in the right temporal for mastication with soft food.

According to the results of this study, we suggest that children with MIH have impairment in crushing and chewing food, a fact that can be proven by the muscle hyperactivity observed. The instability in the biomechanics of mastication may have promoted greater muscle effort, and required greater recruitment of muscle fibers, when compared to children without MIH, causing muscle hyperactivity and decreased masticatory efficiency.

The analysis of skeletal striated musculature is associated with the performance of phonation, mastication, and swallowing activities. The masticatory muscles and their auxiliaries interact in a coordinated way, keeping food on the occlusal surfaces of the teeth to be crushed and subsequently swallowed [62].

To understand the complex functioning of the stomatognathic system, it is of fundamental importance to observe the function of orofacial anatomical structures, in particular the skeletal striated musculature during the recruitment of motor units of muscle fibers for the correct execution of mandibular movement [63].

Dynamic activity is performed by chewing food (soft or consistent) and masticatory efficiency is directly linked to the individual's quality of life, since the entire process of food digestion begins with chewing. Thus, the recovery of masticatory function is one of the main goals to be achieved by dental treatment. An essential tool used for the analysis of masticatory cycles is the envelope integral of the electromyographic signal, capable of analyzing only the periods of isometric contractions and signaling possible changes in masticatory efficiency [64].

For an individual to have a healthy oral health condition, it is necessary that he has a normal occlusion [65], however, the possible destruction of teeth with MIH is evident, and can present from small stains to severe decay of the dental structures [6,63], in which the latter case, the extensive involvement can cause problems in the occlusion and in the static or dynamic activities of the system stomatognathic.

Thus, these data suggest an imbalance in the masticatory functions of children with MIH, and an association with the side of the dental arch most affected by MIH. Therefore, dental treatment for MIH should seek to restore the shape (occlusal anatomy) of the teeth and the function (correct occlusion) to allow the adequate functions of the stomatognathic system to be restored.

When muscle hyperactivity is noticed in the MIHG, regardless of whether the condition is postural or dynamic, it shows there might be an excessive use of a particular muscle to compensate for the difficulty in performing its function, leading to a greater recruitment of muscle fibers to perform the same condition as a healthy muscle [66].

## Conclusions

The results of the evaluation of the stomatognathic system of children with MIH compared with those without MIH indicated muscle hyperactivity in postural and dynamic conditions.

Thus, children with MIH showed impaired stomatognathic system functionality.

## Supporting information

**S1 Raw data.**
(XLSX)

**S1 File.**
(PDF)

## Author Contributions

**Conceptualization:** Milena Rodrigues Carvalho, Simone Cecilio Hallak Regalo, Selma Siéssere, Lígia Maria Napolitano Gonçalves, Francisco Wanderley Garcia de Paula-Silva, Paulo Nelson-Filho, Fabrício Kitazono de Carvalho, Alexandra Mussolino de Queiroz.

**Data curation:** Milena Rodrigues Carvalho, Simone Cecilio Hallak Regalo, Selma Siéssere, Lígia Maria Napolitano Gonçalves, Francisco Wanderley Garcia de Paula-Silva, Fernanda Vicioni-Marques, Paulo Nelson-Filho, Fabrício Kitazono de Carvalho, Alexandra Mussolino de Queiroz.

**Formal analysis:** Milena Rodrigues Carvalho, Simone Cecilio Hallak Regalo, Selma Siéssere, Francisco Wanderley Garcia de Paula-Silva, Fernanda Vicioni-Marques, Paulo Nelson-Filho.

**Funding acquisition:** Milena Rodrigues Carvalho, Francisco Wanderley Garcia de Paula-Silva, Fabrício Kitazono de Carvalho, Alexandra Mussolino de Queiroz.

**Investigation:** Milena Rodrigues Carvalho, Simone Cecilio Hallak Regalo, Selma Siéssere, Lígia Maria Napolitano Gonçalves, Francisco Wanderley Garcia de Paula-Silva, Fernanda Vicioni-Marques, Paulo Nelson-Filho, Fabrício Kitazono de Carvalho, Alexandra Mussolino de Queiroz.

**Methodology:** Milena Rodrigues Carvalho, Simone Cecilio Hallak Regalo, Selma Siéssere, Lígia Maria Napolitano Gonçalves, Francisco Wanderley Garcia de Paula-Silva, Fernanda Vicioni-Marques, Paulo Nelson-Filho, Paulo Batista de Vasconcelos, Fabrício Kitazono de Carvalho, Alexandra Mussolino de Queiroz.

**Project administration:** Milena Rodrigues Carvalho, Simone Cecilio Hallak Regalo, Selma Siéssere, Lígia Maria Napolitano Gonçalves, Fernanda Vicioni-Marques, Fabrício Kitazono de Carvalho, Alexandra Mussolino de Queiroz.

**Resources:** Milena Rodrigues Carvalho, Simone Cecilio Hallak Regalo, Selma Siéssere, Alexandra Mussolino de Queiroz.

**Software:** Milena Rodrigues Carvalho, Simone Cecilio Hallak Regalo, Selma Siéssere, Francisco Wanderley Garcia de Paula-Silva, Paulo Batista de Vasconcelos.

**Supervision:** Milena Rodrigues Carvalho, Simone Cecilio Hallak Regalo, Selma Siéssere, Lígia Maria Napolitano Gonçalves, Paulo Batista de Vasconcelos, Fabrício Kitazono de Carvalho, Alexandra Mussolino de Queiroz.

**Validation:** Milena Rodrigues Carvalho, Simone Cecilio Hallak Regalo, Selma Siéssere, Francisco Wanderley Garcia de Paula-Silva, Fernanda Vicioni-Marques, Fabrício Kitazono de Carvalho, Alexandra Mussolino de Queiroz.

**Visualization:** Milena Rodrigues Carvalho, Simone Cecilio Hallak Regalo, Selma Siéssere, Francisco Wanderley Garcia de Paula-Silva, Fernanda Vicioni-Marques, Fabrício Kitazono de Carvalho, Alexandra Mussolino de Queiroz.

**Writing – original draft:** Milena Rodrigues Carvalho, Simone Cecilio Hallak Regalo, Selma Siéssere, Lígia Maria Napolitano Gonçalves, Francisco Wanderley Garcia de Paula-Silva, Fernanda Vicioni-Marques, Fabrício Kitazono de Carvalho, Alexandra Mussolino de Queiroz.

**Writing – review & editing:** Milena Rodrigues Carvalho, Simone Cecilio Hallak Regalo, Selma Siéssere, Lígia Maria Napolitano Gonçalves, Francisco Wanderley Garcia de Paula-Silva, Fernanda Vicioni-Marques, Paulo Nelson-Filho, Fabrício Kitazono de Carvalho, Alexandra Mussolino de Queiroz.

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
