## [Decision Letter · Decision Letter 0]

17 Aug 2022

PONE-D-22-20949Electromyographic Analysis of the Stomatognathic System of Children with Molar-Incisor HypomineralizationPLOS ONE

Dear Dr. Carvalho,

Thank you for submitting your manuscript to PLOS ONE. After careful consideration, we feel that it has merit but does not fully meet PLOS ONE’s publication criteria as it currently stands. Therefore, we invite you to submit a revised version of the manuscript that addresses the points raised during the review process.

We look forward to receiving your revised manuscript.

Kind regards,

Martina Ferrillo

Academic Editor

PLOS ONE

Journal Requirements:

a) Did participants provide their written or verbal informed consent to participate in this study?

"This research was funded by the National Council for Scientific and Technological

Development – CNPq (Process: 405914/2021-0)."

"This research was funded by the National Council for Scientific and Technological Development – CNPq (Process: 405914/2021-0)."

Reviewers' comments:

Reviewer's Responses to Questions

**Comments to the Author**

1. Is the manuscript technically sound, and do the data support the conclusions?

Reviewer #1: Yes

Reviewer #2: Yes

2. Has the statistical analysis been performed appropriately and rigorously? 

Reviewer #1: Yes

Reviewer #2: Yes

3. Have the authors made all data underlying the findings in their manuscript fully available?

Reviewer #1: Yes

Reviewer #2: Yes

4. Is the manuscript presented in an intelligible fashion and written in standard English?

Reviewer #1: Yes

Reviewer #2: Yes

5. Review Comments to the Author

Reviewer #1: Electromyographic Analysis of the Stomatognathic System of Children with Molar-Incisor Hypomineralization

Dear Authors,

the objective of this study was to conduct a comparative analysis of the electromyographic (EMG) activity of the masseter and temporal muscles (right and left), as well as to evaluate habitual masticatory cycles in individuals with and without Molar-Incisor Hypomineralization (MIH).

The study is in line with the aims of the Journal. Moreover, the study is very interesting and original.

However, the manuscript has to be improved.

Abstract

- The Abract Section is well written. However, please left spaces before and after “=“.

- Moreover, in my opinion the sentence “Conclusions: children with MIH had impaired functionality of the stomatognathic system” is too much hard. Please use “seem to have/or similar” instead of “had”.

Introduction

- The Introduction Section is too short and it do not introduce the electromyography. Please add a section in which you introduce it, reporting how it is used in dentistry (please cite Deregibus A et al. Are occlusal splints effective in reducing myofascial pain in patients with muscle-related temporomandibular disorders? A randomized-controlled trial. doi: 10.5606/tftrd.2021.6615. de Sire A et al. A Telerehabilitation Approach to Chronic Facial Paralysis in the COVID-19 Pandemic Scenario: What Role for Electromyography Assessment?. doi: 10.3390/jpm12030497. Li D et al. Accuracy of sleep bruxism scoring based on electromyography traces of different jaw muscles in individuals with obstructive sleep apnea. doi: 10.5664/jcsm.9940.

- Please modify “[4, 5, 6]” to 4-6

- Please modify “ [7, 8, 9, 10, 11]” to 7-11, for all references in the text.

Material and Methods

“The study included 72 children aged between six and 12 years, divided into the MIH group (MIHG, n = 36) and a control group without MIH (CG, n = 36).” Please put this information in the Result Section.

Discussion

PEB. Please report the definition of peb.

The Discussion section is well written. It reports the recent literature and authors argue their results.

References

The References have to be reported according to the instructions for Authors (https://journals.plos.org/plosone/s/submission-guidelines).

Reviewer #2: Dear Authors,

Your manuscript is really interesting and well conducted.

Unfortunately it cannot be published in present form and it needs to be revised.

• First, i ask you to check the plagiarism of your article using specific sites to get a similitary report (I recommend you grammarly pro: https://www.grammarly.com).

• You need to review the grammar and English of your article, with the help of a native English speaker (you can specify who helped you in reviewing English in the acknowledgements) or simply by using a site that can support you in English

• About the Title of the article, I suggest you to modify it and add the type of article

• Regarding the Abstract you must divide it into sub-paragraphs: Aim-Introduction-Matherials-Results-Conclusions.

• Add recent references about the topic of the article, dwelling in the introduction on articles published in 2022 and describing what your article will add compared to the last articles published; Preferably a published articles should be with 90 or more references, you don’t be afraid to add too many references.

I suggest you some articles that will help you improve your article.

In particular, I suggest you add in your discussion and introduction a few lines about other factors that can alter muscles activity and the chewing for example loss of teeth, decay teeth, mobile deciduous teeth, probably to decrease the risk of bias of your paper could be useful evaluate a DMFT of patients, certainly patients with loss teeth could have an altered muscles activity (compare previous Studies with partial/ total edentulism/syndromes discussing the topic) you can use these Elements to increase your discussion and add the purposes of future studies on the topic

RESULTS OF TREATMENT OF EDENTULOUS PATIENTS WITH DENTURES, MADE OF «VERTEX THERMOSENSE» (THERMOPLASTIC MATERIAL). PMID: 34159933.

Effects of denture wearing on coordinated features of jaw and neck muscle activities during chewing in partially edentulous elderly patients. PMID: 33041278.

Oral-facial-digital syndrome (OFD): 31-year follow-up management and monitoring PMID: 29460530

Telescopic overdenture on natural teeth: Prosthetic rehabilitation on OFD syndromic patient and a review on available literature PubMed ID 29460531

Please expand conclusion section with main results and future perspectives of this study

Thank You,

Kind Regards

6. PLOS authors have the option to publish the peer review history of their article (what does this mean?). If published, this will include your full peer review and any attached files.

Reviewer #1: No

Reviewer #2: No

---

## [Author Response · Author response to Decision Letter 0]

13 Oct 2022

Ribeirão Preto, October 13th, 2022

Editor-in-Chief

Emily Chenette

PlosOne

Manuscript PONE-D-22-20949: Electromyographic Analysis of the Stomatognathic System of Children with Molar-Incisor Hypomineralization

Dear Professor Emily Chenette

 Thank you so much for allowing us to review the manuscript PONE-D-22-20949. In the following, we respond to each of the referee's points and explain the resulting changes in the manuscript. The changes incorporated in the revised manuscript are in bold. The revised manuscript in its submitted form has been read and approved by all authors. 

 Sincerely yours, 

 Professor Carvalho.

Manuscript PONE-D-22-20949: Electromyographic Analysis of the Stomatognathic System of Children with Molar-Incisor Hypomineralization

Dear Dr. Carvalho,

We've checked your submission and before we can proceed, we need you to address the following issues:

1. Re: Table

 Please include a copy of Table 10 which you refer to in your text on page 4.

Answer: The tables were corrected in the text.

2. Can you please upload an additional copy of your revised manuscript that does not contain any tracked changes or highlighting as your main article file. This will be used in the production process if your manuscript is accepted. Please amend the file type for the file showing your changes to Revised Manuscript w/tracked changes. Please follow this link for more information: http://blogs.PLOS.org/everyone/2011/05/10/how-to-submit-your-revised-manuscript/

Answer: The corrections were made in the text.

---

## [Decision Letter · Decision Letter 1]

19 Oct 2022

Dear Dr. Fabrício Kitazono de Carvalho,  

Electromyographic Analysis of the Stomatognathic System of Children with Molar-Incisor Hypomineralization

PONE-D-22-20949R1

We’re pleased to inform you that your manuscript has been judged scientifically suitable for publication and will be formally accepted for publication once it meets all outstanding technical requirements.

Kind regards,

Martina Ferrillo

Academic Editor

PLOS ONE

Reviewers' comments:

Reviewer's Responses to Questions

**Comments to the Author**

1. If the authors have adequately addressed your comments raised in a previous round of review and you feel that this manuscript is now acceptable for publication, you may indicate that here to bypass the “Comments to the Author” section, enter your conflict of interest statement in the “Confidential to Editor” section, and submit your "Accept" recommendation.

Reviewer #2: All comments have been addressed

2. Is the manuscript technically sound, and do the data support the conclusions?

Reviewer #2: Yes

3. Has the statistical analysis been performed appropriately and rigorously? 

Reviewer #2: Yes

4. Have the authors made all data underlying the findings in their manuscript fully available?

Reviewer #2: Yes

5. Is the manuscript presented in an intelligible fashion and written in standard English?

Reviewer #2: Yes

6. Review Comments to the Author

Reviewer #2: Dear authors,

I'm satisfied, and in my opinion the paper is ready for the publication.

I found this work impactful and fit well with in the scope of this journal.

The manuscript needs some minor improvements; there are a few suggestions that authors may consider to improve it further: The use of the English language is reasonable, however, there are a number of punctuation and grammatical errors; that should be corrected and rephrased using academic English for a better flow of text for reader.

Abstract: is precisely written, and the aim of the study is mentioned.

The introduction; is detailed, compact, covering the background information and the rationale of the study effectively.

Materials and methods- this section is well organized.

Result, discussion and conclusion are well written.

Regards

7. PLOS authors have the option to publish the peer review history of their article (what does this mean?). If published, this will include your full peer review and any attached files.

Reviewer #2: No

---

## [Editor Report · Acceptance letter]

15 Feb 2023

PONE-D-22-20949R1 

Electromyographic Analysis of the Stomatognathic System of Children with Molar-Incisor Hypomineralization 

Dear Dr. Carvalho:

I'm pleased to inform you that your manuscript has been deemed suitable for publication in PLOS ONE. Congratulations! Your manuscript is now with our production department. 

Kind regards, 

on behalf of

Dr. Martina Ferrillo 

Academic Editor

PLOS ONE